# Inhibition of Potato *Fusarium* Wilt by *Bacillus subtilis* ZWZ-19 and *Trichoderma asperellum* PT-29: A Comparative Analysis of Non-Targeted Metabolomics

**DOI:** 10.3390/plants13070925

**Published:** 2024-03-22

**Authors:** Jianxiu Hao, Zhen Wang, Yuanzheng Zhao, Shujie Feng, Zining Cui, Yinqiang Zhang, Dong Wang, Hongyou Zhou

**Affiliations:** 1Key Laboratory of Biopesticide Creation and Resource Utilization in Inner Mongolia Autonomous Region, College of Horticulture and Plant Protection, Inner Mongolia Agricultural University, Hohhot 010020, China; haojianxiu@emails.imau.edu.cn (J.H.); cau1022@imau.edu.cn (Z.W.); yq050906@163.com (Y.Z.); 2Guangdong Province Key Laboratory of Microbial Signals and Disease Control, Guangzhou 510642, China; sjief@scau.edu.cn (S.F.); ziningcui@scau.edu.cn (Z.C.); 3Institute of Plant Protection, Inner Mongolia Academy of Agricultural & Animal Husbandry Sciences, Hohhot 010031, China; zhaoyuanzheng.gege@163.com

**Keywords:** potato *Fusarium* wilt, fermentation broth, *Bacillus subtilis* ZWZ-19, *Trichoderma asperellum* PT-29, metabolomics analysis

## Abstract

Potato *Fusarium* Wilt is a soil-borne fungal disease that can seriously harm potatoes throughout their growth period and occurs at different degrees in major potato-producing areas in China. To reduce the use of chemical agents and improve the effect of biocontrol agents, the inhibitory effects of the fermentation broth of *Bacillus subtilis* ZWZ-19 (B) and *Trichoderma asperellum* PT-29 (T) on *Fusarium oxysporum* were compared under single-culture and co-culture conditions. Furthermore, metabolomic analysis of the fermentation broths was conducted. The results showed that the inhibitory effect of the co-culture fermentation broth with an inoculation ratio of 1:1 (B1T1) was better than that of the separately cultured fermentation broths and had the best control effect in a potted experiment. Using LC-MS analysis, 134 metabolites were determined and classified into different types of amino acids. Furthermore, 10 metabolic pathways had the most significant variations, and 12 were related to amino acid metabolism in the KEGG analysis. A correlation analysis of the 79 differential metabolites generated through the comprehensive comparison between B, T, and B1T1 was conducted, and the results showed that highly abundant amino acids in B1T1 were correlated with amino acids in B, but not in T.

## 1. Introduction

Potato (*Solanum tuberosum* L.; family: Solanaceae) is an annual herbaceous plant and the fourth largest food crop in the world, providing various nutrients, such as vitamins, to humans [1]. Potato is the main food crop in China given their high consumption value, with a planting area of 4.915 million hm^2^ and producing 91.881 million t, accounting for 28.3% of the global total planted area and 24.8% of the total production, respectively, in 2019 [2,3]. China currently has the largest potato planting area, yield, and harvest globally [4]. Issues such as single planting techniques, shortened crop rotation cycles, and decreased cultivation levels have disrupted soil microbial communities, leading to increases in various potato diseases, among which *Fusarium* wilt is particularly severe [5,6,7]. Generally, it causes a reduction of about 30% in production and, in some continuous cropping plots, causes more than 78% of plants to die, seriously restricting the healthy and steady development of China’s potato production and processing industry [8]. Potato farmers mostly use chemical agents to control *Fusarium* wilt, but the frequent use of such chemical pesticides negatively affects the environment and food safety [9,10]. Biological control is harmless and long term and is an effective method for preventing and controlling diseases [11]. *Bacillus* spp. exert antagonistic effects against various pathogenic fungi and bacteria [12]. *Trichoderma* is an important biocontrol fungus that can control plant diseases through mechanisms such as competition, the production of antibacterial secondary metabolites, parasitism, and the induction of plant disease resistance [13]. To date, more than 300 secondary metabolites have been isolated from *Trichoderma*, of which more than 45 antimicrobial metabolites have been identified, which are directly related to the efficacy of *Trichoderma* for biological control [14]. However, the use of a single strain often faces issues such as unstable colonisation, whereas the combined use of multiple strains can improve antibacterial activity and enhance their ability to adapt to the environment [15,16,17]. Thus, using multiple microorganisms to jointly control plant diseases can improve the effectiveness of biological control [18]. Wei [19] found that *Bacillus subtilis* and *Chaetomium globosum* had a better control effect on cucumber wilt disease when combined than when applied individually.

Metabolomics technology is used to study metabolites in biological samples, helping to identify and determine mechanisms of action [20,21]. Liquid chromatography–mass spectrometry (LC-MS), gas chromatography–mass spectrometry (GC-MS), and nuclear magnetic resonance have promoted the rapid development of metabolomics [22,23]. For example, metabolomics technology has been used to identify the metabolites, targets, and mechanisms of action of biocontrol bacteria against pathogenic bacteria. Xu et al. (2019) analysed the antifungal activity and functional components of cell-free supernatants from *Bacillus amyloliquefaciens* LZN01, which inhibits *Fusarium oxysporum* f. sp. *niveum* growth [24]. Fan et al. (2022) revealed the biocontrol mechanism of *Purpureocillium lilacinum* by studying its fermentation broth activity and metabolites [25]. Hao et al. (2023) identified 6-PP from a fermentation solution of *Trichoderma asperellum* PT-15 and applied it to the nutrient solution of soilless-cultivated tomatoes to prevent and control tomato *Fusarium* wilt [26]. 

In our previous studies, the volatile compound 2-methylbenzothiazole from *B. subtilis* S-16 had evident inhibitory effects against the mycelium growth of *Sclerotinia sclerotiorum* and *Botrytis cinerea*. A combination ratio of S-16 and PT-29 of 2:1 exhibited the best control against potato *Verticillium* wilt [27,28]. In the present study, a single-culture and co-culture fermentation solution of *B. subtilis* ZWZ-19 and *T. asperellum* PT-29 were used to carry out bacteriostatic tests on their ability to control *Fusarium oxysporum*. We also verified their preventive effect against potato *Fusarium* wilt in potted experiments and carried out a metabolomic analysis using LC-MS/MS. Our findings provided new ideas for the biological control of *Fusarium* wilt.

## 2. Materials and Methods

### 2.1. Pathogens, Biocontrol Strains, and Plants

*Fusarium oxysporum* (GenBank accession: OR523626), which causes potato *Fusarium* wilt, and the biocontrol fungus *Trichoderma asperellum* PT-29 were cultured on Petri dishes containing potato dextrose agar (PDA). The biocontrol bacterium *B. subtilis* ZWZ-19 was cultured in Luria–Bertani (LB) medium. Pathogen and biocontrol strains were obtained from the Fungus Preservation Collection of the Inner Mongolia Agricultural University, China. Susceptible potato plants were provided by the Inner Mongolia Potato Breeding Center, China.

### 2.2. Preparation of the Fermentation Broth and Observation of Morphological Characteristics among Different Culture Methods

Referring to the study by Wu et al. (2018) [15], with minor modifications, the fermentation solution was prepared as follows: *B. subtilis* ZWZ-19 was cultured on LB medium for 2 days, and a single colony was selected for further incubation on LB broth in a shaking incubator at 180 rpm and 28 °C for 24 h until the OD_600_ value was 1.0. Then, 3 mL ZWZ-19 culture fluid was separately inoculated onto 100 mL potato dextrose broth (PDB) and fermented for 3 days at 180 rpm and 28 °C. The resulting fermentation solution named B. *T. asperellum* PT-29 was cultured on PDA and incubated at 28 °C for 3 days. The spores were washed with sterile water, filtered through sterilised absorbent cotton, and inoculated on PDB. The final concentration of spores was adjusted to 10^6^ spores mL^−1^. The samples were then cultured in a shaking incubator at 180 rpm and 28 °C for 24 h until the OD_600_ value was 1.0. Then, 3 mL PT-29 was added to 100 mL PDB and fermented for 3 days at 180 rpm and 28 °C, resulting in a fermentation broth named T. Broth B1T2 contained 2 mL PT-29 and 1 mL ZWZ-19 co-inoculated into 100 mL PDB and fermented at 180 rpm and 28 °C for 3 days. Broth B2T1 contained 2 mL ZWZ-19 and 1 mL PT-29 co-inoculated into 100 mL PDB and fermented at 180 rpm and 28 °C for 3 days. Broth B1T1 contained 1.5 mL PT-29 and 1.5 mL ZWZ-19 co-inoculated into 100 mL PDB medium and fermented at 180 rpm and 28 °C for 3 days. For the control, 3 mL PDB culture solution was inoculated into 100 mL PDB medium and fermented at 180 rpm and 28 °C for 3 days. The morphological characteristics of the different cultivation methods were observed.

### 2.3. Inhibition of F. oxysporum ZY-7 by Fermentation Solution

The aseptic fermentation filtrate was obtained by filtering through a 0.22 μm aseptic filter. Then, 2.5 mL of the filtrate of each group was added to 10 mL of PDA medium cooled to 55 °C in every Petri dish. Subsequently, circular fungal blocks with a diameter of 8 mm were collected using a punch tool and placed at the plate centre for incubation at 25 °C for 3 days. Finally, the colony diameter was measured, and the inhibition rate was calculated as follows:

Inhibition rate (%) = (Control colony diameter − Treatment colony diameter)/Control colony diameter × 100% [29].

Generally, if the inhibition rate exceeds 50%, this indicates a good inhibitory effect [8].

### 2.4. Control Effect of the Fermentation Broth on Potato Fusarium Wilt

The spore concentration of the *F. oxysporum* isolate was adjusted to 1 × 10^6^ spores m^−1^ and was used to inoculate potato plants with five or six leaves planted in pots. And then, the fermentation liquid from each group was poured into the pots. The amount of inoculation and fermentation liquid was 50 mL. Furthermore, the disease index and control effect of potato *Fusarium* wilt were tested after 15 d [30]. The same 6 groups as in Section 2.2 were established, each with three replicates with 10 potato plants per replicate.

SPSS 25.0 software (IBM Corp, Armonk, NY) was used for one-way ANOVA analysis and Graphpad 8.0 (Inc., San Diego, CA, USA) software was used for mapping.

### 2.5. LC-MS/MS Analysis of the Fermentation Broth of T. asperellum PT-29 and B. subtilis ZWZ-19 Co-Culture

According to the test results of Section 2.3 and Section 2.4, the fermentation broths B1T1, B, and T were filtered through a sterilised Millipore filter of pore size 0.22 µm and sent to Beijing Novogene Technology Co., Ltd., Beijing, China for metabolomics testing.

The method proposed by Hu et al. (2023) [31] was slightly modified. All samples were thawed at 4 °C and mixed evenly. Each 1 mL sample was placed in a 1.5 mL centrifuge tube and centrifuged for 10 min at 4500 rpm/min. Then, 200 μL of supernatant was added to 400 μL of extract solution (methanol/acetonitrile = 3:1) and left to stand for 1 h at 4 °C following vortex oscillation. Next, the supernatant was centrifuged at 12,000 rpm/min and 4 °C for 15 min, and then an equal volume of 50% methanol aqueous solution was added. Subsequently, 100 μL of solution containing 50% methanol of each sample was centrifuged for 15 min at 12,000 rpm/min and 4 °C again. Finally, 20 µL of each sample was used for quality control analysis, and the remaining sample was used for LC-MS. 

For chromatographic analysis, the samples were placed in an 8 °C automatic sampler and passed through an ultrahigh performance liquid chromatography system. The HSS T3 1.7 µm (2.1 × 50 mm) chromatographic column was used for gradient elution. The injection volume was 5 μL, the flow rate was 0.4 mL/min, and the column temperature was 40 °C. The chromatographic mobile phases used were as follows: A: 5 mM ammonium formate in water, B: acetonitrile, C: 0.1% formic acid in water, and D: 0.1% formic acid in acetonitrile. The gradient elution procedure was as follows: 0–1 min, D changed linearly from 10% to 30%; 1–19 min, D changed linearly from 30% to 95%; 20–21 min, D changed linearly from 95% to 10%; 21–25 min, D was maintained at 10%. A Thermo QEHF-X instrument was used for electrospray ionisation, and the cation–anion ionisation mode was used for mass spectrometry. The positive and negative ion-spray voltage was 2.50 kV. The sheath gas was 50 arb, and the auxiliary gas was 13 arb. The capillary temperature was 325 °C, the capillary voltage was 35 V/−15 V, and the full scan was conducted at a resolution of 60,000 with a scope of 80–1000 *m*/*z*. CID was applied for the secondary dissociation at a fragmentor voltage of 30 eV [23]. 

### 2.6. Metabolomics Data Analysis

The original data obtained from mass spectrometry were imported into Compound Discoverer 3.1 software for spectral processing and database searching to qualitatively and quantitatively determine metabolites. Quality control was performed on the data to ensure the accuracy and reliability of the results. Next, multivariate statistical analysis was conducted on the metabolites, including principal component analysis (PCA) and partial least squares discriminant analysis (PLS-DA), to reveal the differences in metabolic patterns among the different groups. Hierarchical clustering and metabolite correlation analyses were used to reveal the relationships between different samples [32]. The databases KEGG, HMDB, and LIPID MAPS were used to annotate the identified metabolites and understand their functional properties and classifications.

## 3. Results

### 3.1. Morphological Characteristics of B. subtilis ZWZ-19 and T. asperellum PT-29 and Their Inhibitory Effects on F. oxysporum 

*B. subtilis* ZWZ-19 and *T. asperellum* PT-29 were cultured in different culture styles in different proportions, and their morphological characteristics were observed. Fermentation broth B of ZWZ-19 showed a milky white suspension, whereas fermentation broth T of PT-29 was relatively light and beige, with dense hyphae, after being cultured separately for 3 days. After 3 days of co-cultivation, the fermentation broth B1T1 appeared bright yellow, and the mycelium was clearly visible. The fermentation broth B2T1 turned slightly yellow, indicating that ZWZ-19 was dominant and that a small number of PT-29 mycelia were clustered together (Table 1). B2T1 had a higher inhibitory effect on *F. oxysporum* than T but a lower effect than B. The inhibitory effect of B1T1 was better than that of B, T, B2T1, and B1T2 (Figure 1A,B). The control effects observed in the pot experiment were consistent with these results (Appendix A). Thus, B1T1 has a significantly better inhibitory effect on *F. oxysporum* than other fermentation broths.

### 3.2. Annotation and Multivariate Statistical Analysis of Metabolites 

Through LC-MS/MS analysis, 134 different metabolites were annotated based on the KEGG, HMDB, and LipidMaps databases, among which 47 metabolites were annotated from KEGG and HMDB databases, 14 from HMDB and LipidMaps databases, 3 metabolites from all three databases, 62 metabolites from the HMDB database, and only 4 metabolites from KEGG and LipidMaps databases, respectively, including 30 types of amino acid, 27 types of organoheterocyclic compounds, 18 types of benzenoid, 31 types of lipids and lipid-like molecules, 6 types of nucleosides, 2 types of amines, and 16 other types (Appendix A). Among these metabolites, several substances were reported to inhibit *Fusarium oxysporum* like aconitic acid, alpha amino acid, phenylalanine, hydroxyproline, etc. [33,34,35].

PCA and OPLS-DA can effectively extract the main information from samples and maximise the differentiation between each group of samples to explore differential metabolites [36]. Based on the results in Section 3.1, the following metabolome analysis only focuses on the co-cultured fermentation solution B1T1, which exhibited the best control effect, and the single-culture broths B and T. Differences in metabolite levels between groups were analysed through PCA and OPLS-DA. The results showed significant differences in metabolite components between B and T, B1T1 and B, and B1T1 and T (Figure 2). 

### 3.3. Analysis of Differential Metabolites in Single- and Co-Culture Fermentation Solutions of B. subtilis ZWZ-19 and T. asperellum PT-29

According to LC-MS/MS analysis, differential metabolites were screened by comparing B and T, B1T1 and B, and B1T1 and T based on variable importance projection (VIP) > 1.2 and fold change (FC) ≥3 and ≤0.1. Between B and T, there were 71 differential metabolites, of which 42 were upregulated and 29 were downregulated; between B1T1 and B, there were 90 differential metabolites, of which 58 were upregulated and 34 were downregulated. Between B1T1 and T, there were 34 differential metabolites, of which 20 were upregulated and 14 were downregulated (Table 2).

### 3.4. Comprehensive Comparison of Metabolite Differences

A total of 79 compounds were found to significantly differ among B, T, and B1T1. We found 29 compounds at significantly high proportions in B, 32 in T, and 18 in B1T1 (Figure 3).

### 3.5. Comparison of Amino Acid Production in Fermentation Broths

When *B. subtilis ZWZ-19* and *T. asperellum* PT-29 were co-cultured, there were more types of amino acids in B1T1 than in pure cultures B and T (Table 3). 

### 3.6. KEGG Signal Pathway Analysis of Differential Metabolites

KEGG analysis was conducted on the 79 differential metabolites generated from the comprehensive comparison of B, T, and B1T1. The integrative calculation of impact and −log (p) indicated that the metabolites were involved in 10 metabolic pathways and 3 were related to the metabolism of amino acids, such as alanine, aspartate, and glutamate metabolism. (Appendix A, Figure 4). 

### 3.7. Correlation Analysis of Differential Metabolites 

Correlation analysis of metabolomics can reveal the relationships between metabolites. An increase in the production of one substance can promote or inhibit the production of another. We conducted a correlation analysis of the 79 differential metabolites generated through a comprehensive comparison of B, T, and B1T1. Positive correlations were observed among N-acetyl-L-tyrosine, d-ala-d-alanine, urocanic acid, 2-hydroxyphenylalanine, and Gly-Phe levels in B1T1 cells. 2-hydroxyphenylalanine (B1T1) and LL-2,6-diaminopimelate were abundant in B1B1 and were significantly correlated with indole-3-butyric acid in B. LL-2,6-diaminopimelate (B1T1) was abundant in B1T1 and presented a positive correlation with L-malate in B. There were also positive correlations between 8-aminooctanoic acid and L-malate, between L-methionine sulfoxide and indole-3-butyric, and between Gly-Phea and indole-3-butyric acid in B1T1 and B. The amino acids with abundance in B1T1 had no correlation with amino acids in T. For example, the abundant amino acid N6-Acetyl-L-lysine, LL-2,6-Diaminopimelate in B1T1 were highly correlated with Deoxyguanosine and DL-m-Tyrosine in B respectively.These results showed that the effect on amino acid yield of the co-culture mainly depended on B but not on T (Figure 5). 

## 4. Discussion

Farmers use chemical pesticides as the main control measure for potato wilt disease, but the frequent use of chemical pesticides has a serious impact on the environment and food safety [37]. In our previous studies, we found that the volatile substance 2-methylbenzothiazole in ZWZ-19 has a strong inhibitory effect on *Sclerotinia sclerotiorum* and *Botrytis cinerea* [27]. In addition, the control effect of combining ZWZ-19 and PT-29 at a 2:1 ratio on potato *Verticillium* wilt reached 69.23% [28]. In the present study, we compared the inhibitory effects of *B. subtilis* ZWZ-19 and *T. asperellum* PT-29 on *F. oxysporum* and their control effects on potato wilt under single-culture and co-culture conditions. In addition, we analysed the metabolites of ZWZ-19 and PT-29 under single- and co-culture conditions using LC–MS/MS. The experimental results obtained in this study provide new ideas for the prevention and control of potato wilt disease and provide a theoretical basis for the development of microbial pesticides.

As a broad-spectrum agent, *B. subtilis* in co-culture fermentation broths can control maize sheath blight and strawberry powdery mildew [38,39,40]. However, the control of plant diseases using the co-culture fermentation broth of *T. asperellum* has not been reported to date. Wu et al. found that the antimicrobial effect of a co-culture fermentation broth of *B. amyloliquefaciens* ACCC11060 and *T. asperellum* GDFS1009 was significantly higher than that of pure cultures [41]. In the present study, we found that the co-cultured broth containing *B. subtilis* ZWZ-19 and *T. asperellum* PT29 at a ratio of 2:1 had the best effect on controlling potato wilt disease, with inhibition rates of 68.07% and 76.16%, respectively. These results are consistent with those reported by Wu et al. (2018) [15].

In microbial co-cultures, microorganisms interact with each other and compete for limited nutrients and living spaces. These interactions stimulate the production of bioactive secondary metabolites that cannot be obtained from corresponding pure cultures [15]. In the present study, we annotated 134 substances, 34 of which accounted for 25% of the total. Meanwhile the inhibition rate against *Staphylococcus aureus* and 15 biosynthesised metabolites was detected by optimising the fermentation conditions of a co-culture of *Aspergillus sydowii* and *Bacillus subtilis* in [42]. That study suggested that new antibacterial substances could be discovered by optimising the co-culture conditions of ZWZ-19 and PT29. 

There were more types of amino acids in B1T1 than in B and T. Amino acids have a wide range of applications in food and feed biotechnology and are also used as intermediates in the chemical industry [15]. For examples, Vancomycin is a sugar skin antibiotic that binds to D-Ala-D-Ala to inhibit bacterial cell wall growth, which leads to bacterial cell apoptosis, and Isotretinoin is a type of systemic tretinoin that has significant anti-inflammatory effects [43,44]. A total of 79 metabolites were involved in 26 metabolic pathways, of which 10 pathways were highly variable, and 12 pathways were associated with amino acid metabolism based on the KEGG analysis (Figure 4). Among the 10 metabolic pathways, 5 have been reported to be related to the biological control of pathogens. They participated in the biological control of pathogens through different metabolic pathways. One was involved in cell signal transduction, such as cell wall synthesis, jasmonic acid signal transduction, and auxin regulation, like inositol phosphate metabolism [45]; one induced plant resistance to infection by pathogens, such as Pyruvate metabolism, and three acted through the inhibition of pathogenic bacteria and through competition [46,47,48] (Appendix A). These results were somewhat different to those reported by Wu et al. [15]. Another unreported metabolic pathway is worth studying in the future. Correlation analysis is important to determine the association between different compounds. During production, the yield of one compound can be increased by increasing the amount of the other [49]. Positive correlations were observed between N-acetyl-L-tyrosine, D-Ala-D-Ala, and others. 2-hydroxyphenylalanine and LL-2,6-diaminopimelate were abundant in B1T1 and were significantly correlated with Indole-3-butyric acid in B. LL-2,6-Diaminopimelate was abundant in B1T1, which presented a positive correlation with L-malate in B. Furthermore, this positive correlation occurred between 8-aminooctanoic acid and L-malate, between L-methionine sulfoxide and Indole-3-butyric, and between Gly-Phea and indole-3-butyric acid in B1T1 and B (Figure 5). The amino acids with high yields in B1T1 were not correlated with the amino acids in T. These results showed that the effect on the amino acid yield of the co-culture mainly depended on B, but not on T. 

## 5. Conclusions

In terms of plant disease and pest control, the overuse of chemical pesticides has seriously threatened the environment and food safety. Therefore, it is important to explore environmentally friendly control strategies. Increasing attention has been paid to the use of effective microorganisms to control plant diseases. Some studies have reported that a combination of several biocontrol bacteria is more effective against plant diseases than a single bacterium. The combined use of biocontrol bacteria is a novel strategy for controlling plant diseases. The combined use of biocontrol bacteria involves complex processes. In the future, we will further explore the separation and extraction of active antibacterial ingredients in co-culture fermentation systems and study the differences in metabolic pathways, thereby providing theoretical support for the industrialisation of biocontrol agents.

## Figures and Tables

**Figure 1 plants-13-00925-f001:**
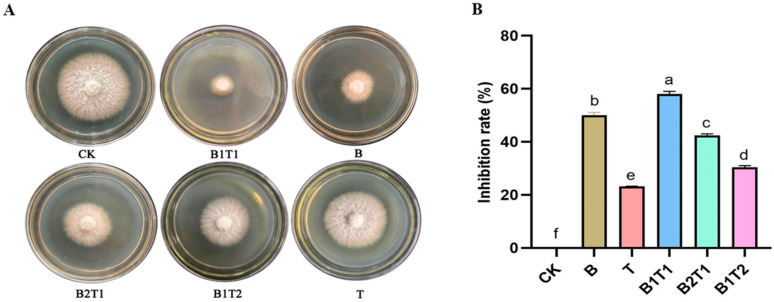
Inhibitory effects of fermentation broths of *B. subtilis* ZWZ-19 and *T. asperellum* PT-29 on *F. oxysporum*. (**A**,**B**) Inhibition effects and rates of fermentation broths on *F. oxysporum* in different culture methods by plate tests and different letters above bars (Tukey’s multiple range test, *p* < 0.05) indicated significant differences.

**Figure 2 plants-13-00925-f002:**
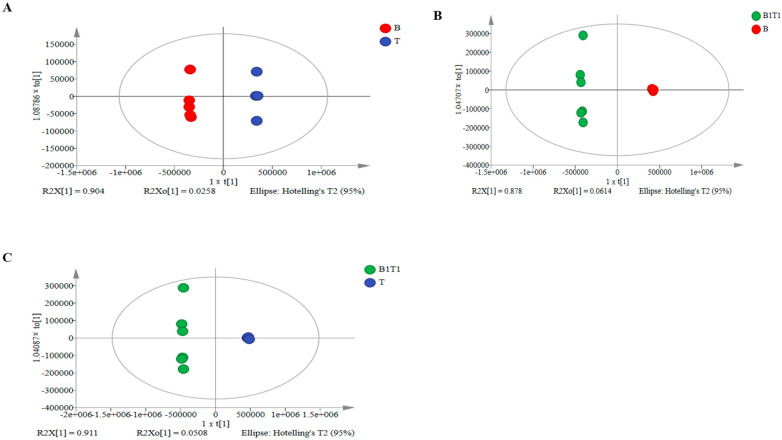
OPLS-DA analysis of the metabolism differences among the fermentation solutions from single-culture and co-culture of *B. subtilis* ZWZ-19 and *T. asperellum* PT-29 based on LC–MS/MS. (**A**) Differences in pure culture fermentation solutions between B and T. (**B**) Differences of fermentation solutions between B1T1 and B, (**C**) Differences of fermentation solutions between B1T1 and T.

**Figure 3 plants-13-00925-f003:**
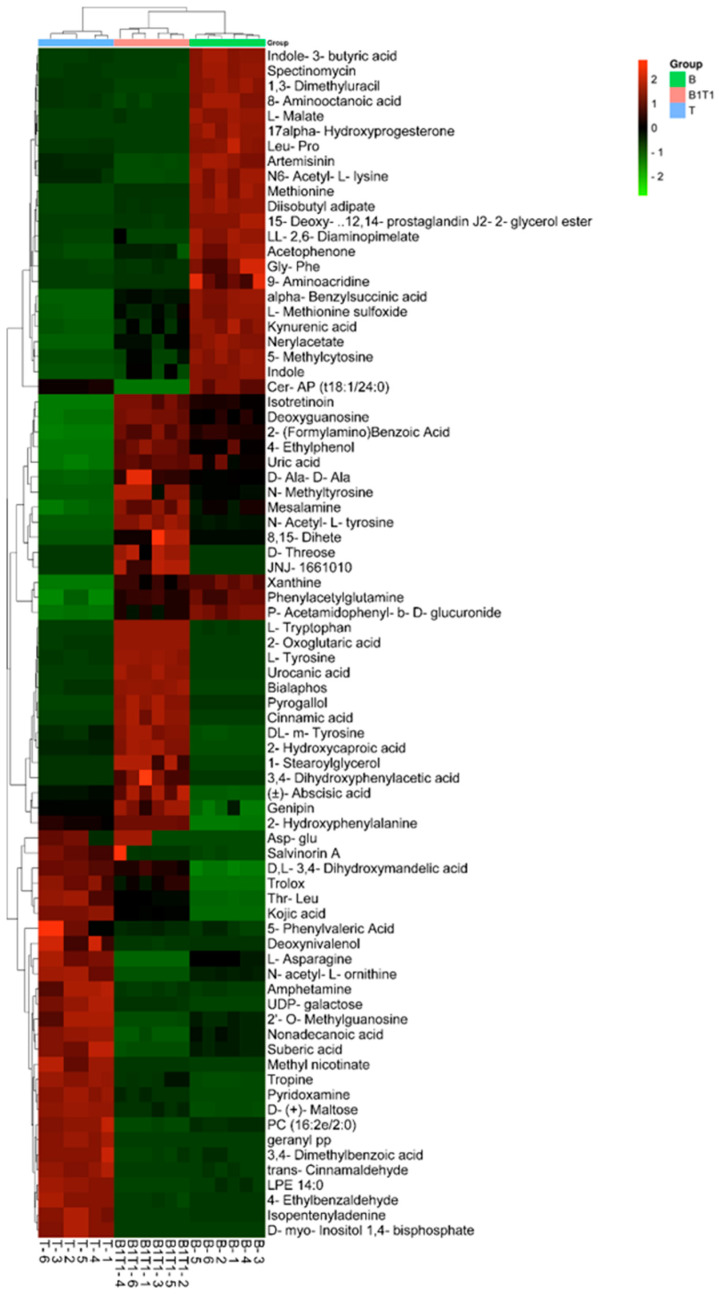
Heatmap of metabolites among the fermentation solutions from pure-cultivation and co-cultivation of *B. subtilis* ZWZ-19 and *T. asperellum* PT-29.

**Figure 4 plants-13-00925-f004:**
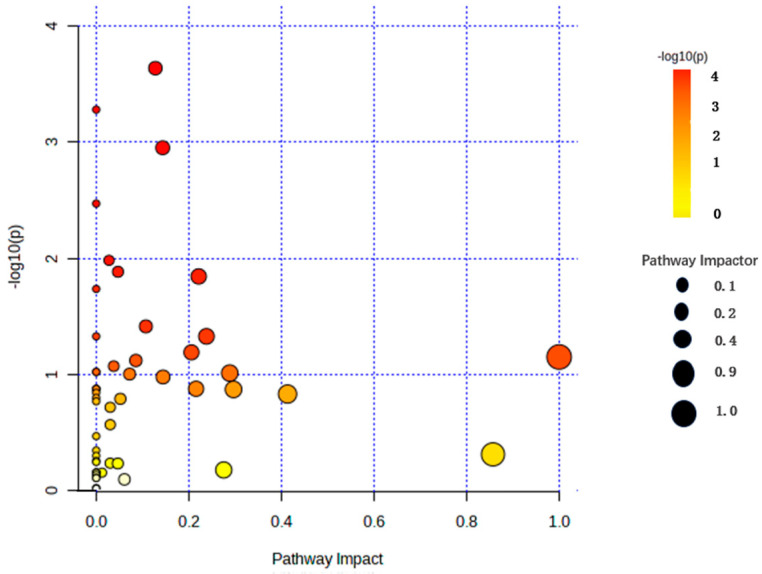
KEGG pathway analysis of the metabolites.

**Figure 5 plants-13-00925-f005:**
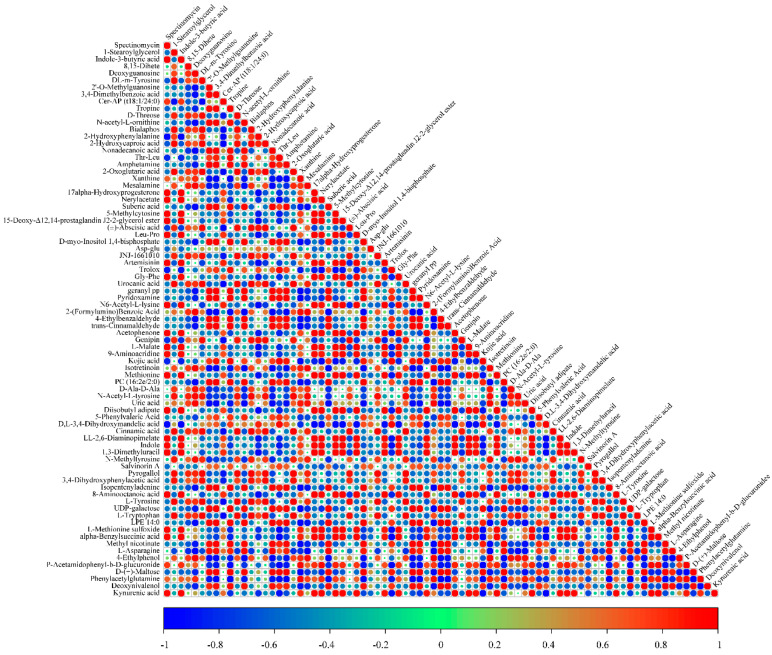
Correlation analysis of metabolites. Red indicates significant positive correlation, blue indicates significant negative correlation, and white indicates no correlation.

**Table 1 plants-13-00925-t001:** Morphology of single-culture and co-culture fermentation broths of *B. subtilis* ZWZ-19 and *T. asperellum PT-29*.

Treatments	Inoculation Amount		Morphology
	(mL)		
	*B. subtilis*	*T. asperellum*	
	ZWZ-19	PT-29	
CK	0	0	None
B	3	0	Milky white suspension
T	0	3	Beige suspension with dense mycelium
B1T1	1.5	1.5	Bright yellow suspension with high viscosity mycelium
B2T1	2	1	Yellowish suspension with small amounts of mycelium
B1T2	1	2	Dark yellow suspension with slightly more mycelium

**Table 2 plants-13-00925-t002:** Differential metabolite comparison between fermentation solutions of single-cultures and co-cultures of *B. subtilis* ZWZ-19 and *T. asperellum* PT-29.

Types	B vs. T			B1T1 vs. B			B1T1 vs. T		
	Total	Upregulation Quantity	Downregulation Quantity	Total	Upregulation Quantity	Downregulation Quantity	Total	Upregulation Quantity	Downregulation Quantity
Organoheterocyclic compounds	20	14	6	22	11	11	3	2	1
Benzenoids	10	4	6	16	14	2	0	0	0
Lipids and lipid-like molecules	16	8	8	20	10	10	10	5	5
Amino acids	12	9	3	16	13	5	17	10	7
Nucleosides	5	4	1	6	3	3	1	1	0
Organic acids	3	1	2	4	2	2	1	0	1
Others	5	2	3	6	5	1	2	2	0
Total	71	42	29	90	58	34	34	20	14

**Table 3 plants-13-00925-t003:** Amino acid production under different cultivation types.

Cultivation	Compound	VIP	*p*-Value
B	L-Malate	1.39 × 10^−8^	1.58013773
B	1,3-Dimethyluracil	1.25 × 10^−6^	1.546674027
B	Phenylacetylglutamine	6.69 × 10^−6^	1.505866262
B	L-Methionine sulfoxide	8.67 × 10^−4^	1.528419213
B	Kojic acid	1.83 × 10^−3^	1.577939364
B	L-Asparagine	8.65 × 10^−3^	1.523708247
B1T1	N-Acetyl-L-tyrosine	2.86 × 10^−7^	1.570824956
B1T1	D-Ala-D-Ala	2.15 × 10^−5^	1.571912954
B1T1	2-Hydroxyphenylalanine	1.12 × 10^−5^	1.667481741
B1T1	L-Tyrosine	1.61 × 10^−4^	1.539024139
B1T1	LL-2,6-Diaminopimelate	1.90 × 10^−4^	1.560211492
B1T1	Isotretinoin	1.93 × 10^−4^	1.577901393
B1T1	8-Aminooctanoic acid	1.16 × 10^−3^	1.539511272
B1T1	Urocanic acid	1.97 × 10^−3^	1.606733748
B1T1	Gly-Phe	3.15 × 10^−3^	1.607442416
T	Asp-glu	6.00 × 10^−6^	1.611481292
T	Thr-Leu	2.14 × 10^−4^	1.666121562
T	N-acetyl-L-ornithine	2.69 × 10^−4^	1.677736612
T	Pyridoxamine	2.35 × 10^−3^	1.599751113
T	Nicotinuric Acid	3.47 × 10^−3^	1.484749249

## Data Availability

The data supporting this study can be accessed by contacting the first author by e-mail at haojianxiu1003@163.com.

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
