# Peer review of "Inhibition of Potato Fusarium Wilt by Bacillus subtilis ZWZ-19 and Trichoderma asperellum PT-29: A Comparative Analysis of Non-Targeted Metabolomics"

_plants, 2024, doi:10.3390/plants13070925_

Round 1

Reviewer 1 Report

Comments and Suggestions for Authors

 The work submitted for review is coherent and well-written. It has minor shortcomings, which are listed below.

 „co-culture broth” should be changed.

„Our findings lay a theoretical foundation for the development of composite biocontrol agents.” This sentence is a little controversial.

Figure 1 should be unified.

 Most of the work is focused on metabolites. However, perhaps, a correction of the title should be considered, because Fusarium inhibition itself appears to be a side element.

I wish the authors success in their continued research.

Reviewer 2 Report

Comments and Suggestions for Authors

The study Potato Fusarium Wilt Inhibition by Different Culture Methods of Bacillus subtilis ZWZ-19 and Trichoderma asperellum PT-29 and Comparative Analysis of Non- targeted Metabolomicsinvestigates the potential of Bacillus subtilis ZWZ-19 and Trichoderma asperellum PT-29 for controlling Fusarium oxysporum, a pathogen causing potato Fusarium wilt. The authors observed morphological characteristics and inhibitory effects of the microorganisms, finding that a co-culture (B1T1) showed the best control effect. Metabolomic analysis revealed significant differences in metabolite profiles between the co-culture and single cultures, particularly in amino acid production. Pathway analysis highlighted metabolic pathways related to amino acid metabolism. Overall, the study suggests that the combined use of B. subtilis ZWZ-19 and T. asperellum PT-29 could be an effective biocontrol strategy for potato Fusarium wilt. Specific comments below:

  1. The title of the manuscript is bit complicated and confusing, it should be concise and direct. Consider rephrasing it to something like "Biological Control of Potato Fusarium Wilt: Bacillus subtilis ZWZ-19 and Trichoderma asperellum PT-29 Culture Methods and Metabolomic Analysis." or "Inhibition of Potato Fusarium Wilt by Bacillus subtilis ZWZ-19 and Trichoderma asperellum PT-29: A Comparative Analysis of Non-targeted Metabolomics" or what authors think is appropriate.

  2. It would be better to understand the significance of the study if the authors included statistics on potato production in China and the economic losses caused by Fusarium wilt.

  3. “Targeted or untargeted metabolomics technology has been widely used for studying metabolites in biological samples, including for identification and determination of mechanisms of action" could be revised to "Metabolomics technology is used to study metabolites in biological samples, helping to identify and determine mechanisms of

    action”.

  4. Specifically mention the methods used for assessing the inhibitory effects on F.

    oxysporum. What criteria were used to determine the level of inhibition against Fusarium

    oxysporum.

  5. Does the authors have the exact number of metabolites annotated from each database

    (KEGG, HMDB, LipidMaps).

  6. Authors have specified the types of amino acids that were more abundant in B1T1

    compared to B and T. However, how do these differences contribute to the biocontrol

    efficacy of the co-culture has not been discussed.

  7. Provide more details on the 10 metabolic pathways identified and their relevance to the

    study. How do these pathways relate to the biocontrol mechanisms of B. subtilis ZWZ-19 and T. asperellum PT-29.

Comments on the Quality of English Language

Overall, the article is well written with no major problem in english language. However, some sentences should be rephrased and rewritten for better readability. Also, authors should ensure consistency in verb tenses throughout the article. For example, in the sentence "Bacillus subtilis ZWZ-19 and Trichoderma asperellum PT-29 were cultured in different culture styles in different proportions," the verb "were cultured" should be in the past tense to match the context. Authors should proofread the manuscript thoroughly before submitting again. 

Reviewer 3 Report

Comments and Suggestions for Authors

#The MS presents a comprehensive study on the inhibitory effects of the fermentation broth of Bacillus subtilis ZWZ-19 (B) and Trichoderma asperellum PT-29 (T) on Fusarium oxysporum, a soil-borne fungal disease affecting potatoes. The study compares the single culture and co-culture conditions of these biocontrol agents, emphasizing the goal of reducing chemical agent use.

#The clarity and organization of the text are commendable.

#The introduction effectively highlights the importance of addressing Potato Fusarium Wilt and sets the context for the study.

#The methodology is well-described, specifying the inoculation ratio and the comparison of fermentation broths under different conditions.

#The findings are presented logically, starting with comparing inhibitory effects and subsequently delving into metabolomic analysis results.

#The use of LC-MS analysis to identify and classify 134 metabolites, especially focusing on amino acids, demonstrates a robust analytical approach.

#Additionally, identifying 10 metabolic pathways with significant variations and their relevance to amino acid metabolism adds depth to the study.

# However, some areas could benefit from further clarification or expansion.

#For instance, the text mentions the best control effect in a potted experiment. Still, it does not provide specific details on the experiment, such as the experimental setup, duration, or key observations.

#In the metabolomic analysis section, while the number of determined metabolites is given, it would be helpful to know whether any of these metabolites are specifically linked to the inhibition of Fusarium oxysporum or if they have broader implications in potato health.

#The correlation analysis is a valuable addition, shedding light on the relationship between differential metabolites in B, T, and B1T1. However, it would enhance the clarity if specific examples of highly abundant amino acids and their correlation patterns were provided.

#The text effectively communicates the study's objectives, methodologies, and findings. Addressing the above-mentioned points would further enhance the clarity and depth of the research presentation.
